# Polish Court Ruling Classification Using Deep Neural Networks

**DOI:** 10.3390/s22062137

**Published:** 2022-03-09

**Authors:** Łukasz Kostrzewa, Robert Nowak

**Affiliations:** Institute of Computer Science, Warsaw University of Technology, Nowowiejska 15/19, 00-665 Warsaw, Poland; lukasz.kostrzewa.stud@pw.edu.pl

**Keywords:** law text classification, machine learning, natural language processing, artificial neural networks, Polish court rulings

## Abstract

In this work, the problem of classifying Polish court rulings based on their text is presented. We use natural language processing methods and classifiers based on convolutional and recurrent neural networks. We prepared a dataset of 144,784 authentic, anonymized Polish court rulings. We analyze various general language embedding matrices and multiple neural network architectures with different parameters. Results show that such models can classify documents with very high accuracy (>99%). We also include an analysis of wrongly predicted examples. Performance analysis shows that our method is fast and could be used in practice on typical server hardware with 2 Processors (Central Processing Units, CPUs) or with a CPU and a Graphics processing unit (GPU).

## 1. Introduction

Processing legal documents such as court rulings is a complex task. Such documents can be several pages long and are less structured than invoices, orders, or shipping documents. Moreover, they use abstract legal concepts that are difficult to learn on a word-by-word basis. In addition, such texts often have specific vocabulary and syntax.

In recent years, we have seen significant progress in the development of Artificial Intelligence (AI) methods. Researchers have begun to implement such methods to support lawyers. One task where AI can be used successfully is the classification of legal texts, namely, the process of assigning a predefined class label to a legal document.

### 1.1. Numerical Representation of Words

Machine learning models can process text paragraphs if the words are transformed to a numerical format. The common approach is to represent each word as an N-dimensional vector. Such a representation allows one to achieve two goals: (1) Synonyms—words with similar meanings should have similar numerical representations. (2) Analogy—words should be identified based on their relationship. For pairs of words representing a relation, such as a→b (e.g., “Poland” → “Warsaw”) and c→d (e.g., “France” → “Paris”), it is expected that $\vec{b}-\vec{a}+\vec{c}$ should be close to $\vec{d}$, where $\vec{a}$ is a vector representation of the word *a*, etc.

Generating a high-quality vector representation of words is a complex task. One of the most common methods is using *Word2Vec* [1]. This method is based on the assumption that the word can be deduced from its context. The *Word2Vec* approach uses neural networks with Continuous Bag-of-Words or Skip-Gram architectures. In Continuous Bag-of-Words, the word is predicted from its neighboring words. Skip-Gram uses the opposite approach—it predicts the word’s context given the word itself. Both models are based on the conditional probability of a word occurring in the context of another word. This probability is formulated as a softmax function, which has the disadvantage of a high computational cost. However, there are computationally efficient approximations of the softmax function. The most popular of these are Hierarchical Softmax and Negative Sampling. Hierarchical Softmax uses a binary tree to represent the vocabulary, in which words are represented as leaves. Conditional probability can be computed by traversing the tree from the root node to the given leaf. Negative Sampling simplifies the problem of distinguishing a given word from all other words in the vocabulary to distinguishing it from draws from the noise distribution [2].

### 1.2. Polish Language Embeddings

The Polish language largely uses inflection to express syntactic relations within a sentence. For such languages, embeddings should include various orthographic forms of words that indicate, among others cases, gender and number.

Vector representation of Polish words has been analyzed by Mykowiecka et al. [3]. These authors trained models using Polish Wikipedia and the National Corpus of Polish corpora. Corpora datasets were divided into lemmas generated by a tagger and orthographic forms. For some models, only some parts of speech were selected. The rest used all the words from the corpora. CBOW and Skip-Gram architectures were used with 100- and 300-dimensional vectors. For learning, the authors used Hierarchical Softmax and Negative Sampling algorithms [4]. Both synonyms and analogies were used to compare models. The paper concluded that it was impossible to select a single best model. However, only models trained on corpora with orthographic forms can be used for ruling classification. For such models, Skip-Gram decisively outperforms CBOW. In addition, Skip-Gram trained using the Negative Sampling algorithm was slightly better than that trained using Hierarchical Softmax.

### 1.3. Machine Learning Models

The recent development of deep learning had a significant impact on Natural Language Processing (NLP). Neural Network models are now able to solve common NLP tasks much better than state-of-the-art traditional methods and require far less manual work. In NLP, the spatial relationship between words is important. Models that are commonly used include Convolutional and Recurrent Neural Networks [5].

#### 1.3.1. Convolutional Neural Networks (CNNs)

Convolutaional neural networks (CNNs) were introduced by Lecun Y. et al. [6] and were originally invented for computer vision. A CNN utilizes the internal structure of data. It consists of computation units that take only a small part of the input vector as their input (e.g., a small part of the image). This approach allows a CNN to extract low-level features in the initial layer and transform them to more high-level ones in the following layers. CNNs have been shown to be efficient in various NLP tasks, e.g., text classification [7,8] and machine translation [9]. Their architecture allows one to utilize spatial relationships between words—to process words together with their context.

#### 1.3.2. Recurrent Neural Networks (RNNs)

Recurrent neural networks (RNNs) are similar to standard feed-forward networks. In their simplest version, to compute the i-th hidden state, they use the i-th input and the (i − 1)th hidden state. RNNs suffer from two problems—unstable gradients and a relatively quick loss of information. The gradients in RNNs tend to vanish or explode. RNNs call the cell function multiple times (once for each step), which leads to strong nonlinear behavior [10]. There are various ways to mitigate both problems, such as clipping the gradient [11], introducing leaky units, and using gated cells—long short-term memory or gated recurrent unit.

Long-Short Term Memory (LSTM) cells, presented by Hochreiter S. et al. [12], introduce self-loop (memory cell) and gates. A memory cell allows the neural network to store a long-term state, and gates allow one to filter data. LSTM can detect long-term dependencies and converge faster, so it has been used successfully in processing long text, time series, audio recordings, etc. [5].

A Gated Recurrent Unit (GRU) is a simplified version of the LSTM cell [13]. They lack memory cells and have fewer gates.

#### Legal Document Classification

Legal document classification has not been studied very widely according to the available literature. However, in recent years, a few successful machine learning implementations in a legal domain have been reported.

French Supreme Court rulings were the subject of a study by Sulea O. et al. [14]. An ensemble of Support Vector Machine (SVM) models was used to classify eight areas of law with 96.8% accuracy and a 96.5% $F_{1}$-score.

Undavia S. et al. classified English Supreme Court opinions into 15 categories [15]. Various embeddings (Word2Vec, GloVe, fastText) and classifiers were examined: CNN, LSTM and GRU neural networks, SVM, and Logistic Regression. The best accuracy (72.4%) was achieved with the Word2Vec and CNN models.

Fernandes W. et al. classified Brazilian Appellate Court decisions by identifying modifications of lower court decisions proposed by the upper court [16]. The corpus consisted of 3022 documents divided into six categories. Embeddings were trained separately, using the Word2Vec model and a dataset of 92,122 appeal decisions. Five models were tested: Bidirectional LSTM (BiLSTM) and GRU (BiGRU), Conditional Random Fields (CRFs), a combination of BiLSTM and CRFs, and a combination of BiGRU and CRFs. The best results were obtained for the combined BiLSTM and CRF model with an $F_{1}$-score of 94.5%.

The use of Artificial Intelligence methods has not been limited to court ruling classification. Lulu W. et al. classified relatively long documents from the U.S. Securities and Exchange Commission [17]. The documents’ text was split into smaller segments and concatenated using a Bidirectional LSTM layer. For classification, a linear classifier, or SVM, was used. SVM classifiers achieved the best results (98.11% accuracy) with a 7-segment split.

Waltl B et al. classified German legal norms into nine categories [18]. From the German Civil Code, 601 sentences were selected that constitute the tenancy law. They were divided into nine semantic types, including Duty, Indemnity, and Permission. The class imbalance was significant—only eight occurrences of Indemnity were found, while Permission appeared 148 times. There were three pre-processing types: none, stop-words removal, and stop-words removal with Term Frequency–Inverse Document Frequency (TF-IDF) transform. Five classifiers were trained: Naive Bayes, Logistic Regression, SVM with linear kernel, Random Forest, and Multilayer Perceptrons. For each model, the best result was achieved with no pre-processing. After 10-fold cross-validation, the best average $F_{1}$-scores were obtained for SVM, 0.83 ± 0.08, and Logistic Regression, 0.81 ± 0.09. Only these two models performed better than rule-based classification.

Mariana Y. N. et al. analyzed the classification of Petitions to Public Prosecution Service in Portuguese [19]. The aim was to classify 18 areas of law (civil, child and youth, criminal, consumer, etc.). They studied various numerical representations of words (TF-IDF as well as Generic and Specialized Embeddings) and different machine learning models (Logistic Regression, SVM, Random Forest, Gradient Boosting, CNN, LSTM, and GRU Neural Networks). The best results—90% accuracy and 85%
$F_{1}$-score—were achieved with the LSTM model. The best word embeddings were prepared using the *Word2Vec* [1] model trained on a domain-specific corpus.

Ruggeri F et al. studied unfairness detection in consumer contracts [20]. Their data consisted of 21,063 sentences from 100 Terms of Service documents from online platforms, and 2346 sentences were labeled as containing an unfair clause. Such clauses were divided into five categories, e.g., limitation of liability or content removal. For each category, binary classification was studied using SVM, CNN, LSTM, and Memory-Augmented Neural Networks (MANNs) with weak (WS) or strong (SS) supervision. The best mean $F_{1}$-score on 10-fold cross-validation was obtained by MANN SS (0.526–0.666, depending on the category) for four categories and SVM (0.673) for the fifth category.

#### Main Objectives of the Presented Work

Research in legal document classification has been investigated for various languages, e.g., English, French, Portuguese, or German. To the best of our knowledge, no successful implementation exists for the Polish language. Our paper presents methods for the classification of Polish court rulings. We use authentic, anonymized Polish court rulings from 1998 to 2020. We analyze various general language embeddings and multiple neural network architectures, focusing on simple Convolutional and Recurrent Neural Networks. The results show that such models can classify documents with very high accuracy (>99%).

We also analyze feature engineering methods to extract the references to legal norms. Such references are grouped into categories based on domain knowledge and are then used as additional input for the model. We observe that feature engineering does not improve model accuracy significantly, but it accelerates the training process.

## 2. Materials and Methods

### 2.1. Polish Court Ruling Dataset

We acquired data from the website maintained by the Polish Ministry of Justice. This data is an anonymized set of court rulings freely available to researchers. The results were processed using the Beautiful Soup tool (https://www.crummy.com/software/BeautifulSoup/ accessed on 15 January 2022). Each record consists of the ruling text and a few metrics. Metrics statistics are presented in Table 1.

We wanted to categorize rulings by the type of case (whether it is a civil case, criminal, etc.). To label the data, we used the Department feature. Information about the department is present in all rulings (Table 1). In the dataset, there were 136 unique departments. Many departments had an ordinal number, e.g., I Civil Department or II Civil Department. Sometimes a place of the department also appeared in the name, e.g., VII Local Criminal Department in Kolno. Sometimes, there was a separate department for appeal cases, e.g., Civil Appeal Department. In 1162 records, there was a typo in the department name. After removing information about the ordinal number, location, and appeal and fixing typos, the number of departments was reduced to 10. We had two additional issues with those departments. Firstly, there was a big difference in the number of rulings, e.g., Community Trade Marks Court had only 66 cases, Business Bankruptcy just 6, while Labor had 34,311 and Civil had 66,877. Secondly, smaller courts often do not have specialized departments; e.g., family-related cases are processed by the Civil department. For rulings with the topic Alimony, 406 cases were processed in the Family department, 345 in the Civil department, and 272 in the Civil Family department. Hence, it is reasonable to merge such departments into one. We merged Family Department and Civil Family Departments into Civil Department, Competition Protection Court, Community Trade Marks Court, and Business Bankruptcy into Economic Department, and Penitentiary into Criminal Department. Eventually, we had four classes: Civil, Economic, Criminal, and Labor. The class imbalance was not high (Figure 1).

Ruling text required pre-processing. We removed all records without the text of the ruling. We replaced all-white spaces with a single space and deleted non-alpha-numerical characters. We removed HyperText Markup Language (HTML) tags. Most of the court rulings began with an introduction, which contained the name of the court department. We deleted introductions, as they could greatly facilitate classification. A similar approach was applied by Sulea O. et al. [14]. After pre-processing, our dataset consisted of 144,784 rulings. The length of a ruling’s text varies from 69 to 206,385 words. Its distribution is presented in Figure 2.

We used feature engineering to create additional input for ruling processing—information about legal references in the ruling text. Legal references are often abbreviated, e.g., art. 477 § 1 k.p.c. (article 477, paragraph 1, Civil Procedure Code). Automatic extraction of such abbreviations from the text would be difficult. Fortunately, most legal references were marked as a link to the given legal act. Downloaded, raw data also contained HTML tags. Legal references in the ruling text could be easily extracted from the raw data by searching for <a> tags used in HTML to represent a link. For each ruling, a list of all legal references was extracted. The total number of references was 2,581,692. On average, 17.83 references per ruling. We divided legal references into 15 categories: 14 legal codes and one category for other legal acts. Legal codes are special statuses that exhaustively cover a particular area of law. They are the most commonly referenced legal norms in rulings. Important codes include the Civil Code, Criminal Code, and Labor Code. We counted the number of references in the given category for each ruling and saved that information as additional features.

The processed text of each ruling was stored in a separate text file. Files were stored in four directories indicating the class. Other metrics and legal references were stored in two CSV files.

We split the whole dataset into training and test subsets. We used 80% of the rulings (115,829) for training and the remaining 28,955 for testing. The division was random and stratified by keeping the identical class proportions in both subsets. The test subset was only used to evaluate the final models’ performance.

### 2.2. Models for Polish Court Rulings Classification

We analyzed various RNNs and CNNs (Table 2). First, we checked various embedding matrices using a baseline model (lstm-dense-*). After selecting the best embeddings, we then checked different architectures.

#### 2.2.1. Baseline Model

We used a simple Recurrent Neural Network with LSTM cells as a baseline model. According to [21], simple RNNs should be sufficient in document classification tasks. Details of the base model (layers, parameters, etc.) are depicted in Figure 3. We used the baseline model to study various embedding matrices (different training methods and vector sizes) and compare the quality and training time of other models.

#### 2.2.2. Convolutional Models

We tested three convolutional models:*conv-max-dense*—presented in Figure 4;*conv-avg-dense*—the same as the previous, but with average pooling;*conv*—a simple convolutional model with average pooling and no hidden dense layer.

Models are summarized in Table 2.

#### 2.2.3. Recurrent Models

Besides the baseline model with different embedding matrices, we tested a few other recurrent neural networks:*gru-dense*—the same as the baseline model, but with GRU cells;*lstm and gru*—simple recurrent models without a hidden dense layer and without LSTM or GRU cells (the latter is presented in Figure 5);*bd-gru-dense*—a more complex architecture with a bidirectional recurrent layer (presented in Figure 6).

The details are depicted in Table 2.

#### 2.2.4. Law References Models

A ruling often references legal documents. The vast majority of these references are to legal codes. Law references were included in the *gru-dense-law* model. Fifteen features (integer numbers) represent such references: 14 legal codes and ’other legal documents’ (15th feature). Each integer is the number of references to the given legal document in the ruling. Recurrent and dense layers are used to process the text input and a dense layer to process law references (Figure 7). The output is then concatenated and passed to the output layer.

### 2.3. Software Implementation

The whole pipeline was implemented in Python 3.8.5. The core logic is a *rulings* Python package. Additionally, we developed various scripts, utility functions, and Jupyter notebooks to process and analyze data and models.

We implemented models using tf.keras.Model (https://www.tensorflow.org/api_docs/python/tf/keras/Model accessed on 15 January 2022). During training, we read data from disk into tf.data.Dataset, which iterates files in a stream fashion and has no requirements to store the full dataset in memory. We used embedding matrices [4]. The matrices were generated using the Gensim library. We used gensim.models.KeyedVectors (https://radimrehurek.com/gensim_3.8.3/models/keyedvectors.html accessed on 15 January 2022) to read and use embeddings.

### 2.4. Training Process

As an optimization algorithm, we chose Adam, which is a gradient-based stochastic algorithm [22]. As a loss function, we used categorical cross-entropy. Both Adam and cross-entropy are standard solutions in training neural networks and are implemented in popular machine learning libraries. Our initial results showed that they give good enough results in our problem. We also prevented over-fitting by applying early stopping. It is a regularization technique that stops training when model parameter updates no longer improve performance on a validation set. We used five-fold cross-validation to better evaluate various models. Data presented in the Results section is an average of five folds. Data used for model selection consisted of 115,829 rulings. In each fold, 80% of it was used for training (92,663 rulings) and 20% for validation (23,166 rulings). Python scripts were run using Python 3.8.5 on a virtual machine with 8 Virtual Central Processing Units (VCPUs), 16GB RAM, and an Ubuntu Server 20.04 operating system.

### 2.5. Metrics

To evaluate models, we used mostly accuracy and $F_{1}$-score, $F_{1}=2\frac{precision*recall}{precision+recall}$. We computed precision, recall, and $F_{1}$-score for each class separately using a one-vs-all strategy. We then calculated the mean values from all four classes.

## 3. Results

### 3.1. Embedding Matrix Selection

We selected four embedding matrices for comparison. We used matrices prepared by Mykowiecka et al. [3]. All matrices were trained using Polish Wikipedia and the National Polish Language Corpora and included all figures of speech and orthographic forms. The embedding vector size was 300. Matrices were trained using CBOW or Skip-Gram models and Negative Sampling (NS) or Hierarchical Softmax (HS) algorithms. Our base model (in Figure 3) was trained using these four matrices. Table 3 presents mean metrics on the validation set, and Figure 8 presents the training history for one of the folds. Embedding matrices trained using Skip-Gram significantly outperformed CBOW. Skip-Gram trained using Negative Sampling was slightly better than that trained using Hierarchical Softmax. It was therefore selected for further optimization. These results are consistent with the findings by Mykowiecka et al. [3], where Skip-Gram trained using Negative Sampling was also the best among models trained on orthographic forms.

### 3.2. Model Selection

Table 4 shows metrics for various neural network architectures. Unless otherwise stated, models use embeddings trained on Skip-Gram with a Negative Sampling model, and the input text sequence length is 300.

### 3.3. Legal Document Reference Model Analysis

The model with legal references *gru-dense-law* has just a slightly better average performance than the similar model with text input only—*gru-dense*. The difference is so small that we should analyze the distribution of accuracy from five folds. Model *gru-dense-law* has an accuracy of 0.9940 ± 0.0007, and *gru-dense* has one of 0.9939 ± 0.0003 (the second number is the standard deviation). Hence, there is no statistical difference between the models. However, the model with legal references requires, on average, fewer epochs to coverage—9.6 compared to 11.8 in the *gru-dense* model. The training process is about 22% faster. Legal references do not improve the model’s accuracy, but they speed up the training process.

### 3.4. Hyper-Parameter Optimization

Models *conv-max-dense*, *gru*, and *bg-gru-dense* were selected for hyper-parameter optimization, as they achieve good results and have different architectures.

For the *conv-max-dense* network, the following parameters were tuned: the filter number (**32**, 64, and 128) and kernel size (3, 4, and **5**) in the convolutional layer, the filter number in the dense layer (**32**, 64, and 128), the pool size in the pooling layer (3, 4, and **5**), and the input text sequence length (300, **400**, and 500). The best set of parameters is in bold.

For the *gru* network, various filters in the recurrent layer were tested (32 and **64**). Moreover, we tested the masking of zero values in the embedding layer (no masking and **with masking**).

For the *bg-gru-dense* network, the following parameters were tested: the filter number in the recurrent layer (32 and **64**), the filter number in the dense layer (**32** and 64), and the masking of zero values in the embedding layer (no masking and **with masking**).

Table 5 shows metrics for the selected models. Metrics of all three have improved after parameter tuning. Model *gru* slightly outperformed *bd-gru-dense*.

### 3.5. Final Model Quality

After selecting the optimal values of parameters depicted in the previous section, models were trained on the whole training set. The average number of epochs from cross-validation was chosen as the number of epochs to train the models. Models were validated on a test set that was not used during the training process and contained 28,955 rulings. The number of rulings in each department is presented in Figure 9.

Table 6 presents metrics on the test set. All three models achieved a good performance. Their metrics were only slightly worse than they were on the validation data. Table 7, Table 8 and Table 9 present confusion matrices for the predictions of the test set.

#### 3.5.1. Performance Analysis

Resource utilization analysis was performed using two Python profilers: (1) memory-profiler (https://pypi.org/project/memory-profiler/ accessed on 15 January 2022), to analyze memory usage, and (2) line-profiler (https://pypi.org/project/line-profiler/ accessed on 15 January 2022), to analyse training and prediction times. We evaluated the training and prediction time for two configurations: the CPU and the GPU. The computer with the CPU was a virtual machine with 8 VCPUs and 16GB RAM running on an Intel Xeon E3 v2 3.50 GHz with eight cores. The machine with the GPU was a virtual machine with 4 VCPUs and 16GB RAM with an NVIDIA GeForce GTX 1080 Ti graphics card. The GPU was accessed using PCI passthrough. Table 10 and Table 11 present the memory usage and training time. Each experiment was performed three times on the same data. In the tables, the average values and standard deviations are presented.

Table 12 and Table 13 present the memory usage and prediction time. The full pipeline includes text pre-processing and neural network model loading. Each measurement was performed three times on the same data. Average values and standard deviations are shown in the tables.

In one of the CNN models, i.e., *conv-max-dense*, a significant speedup is observed. The training time on the GPU is about 4.5 times smaller than on the CPU.

Training time for RNN models—*gru* and *bd-gru-dense* is much longer on the GPU than on the CPU. Both models are relatively small in terms of the number of trainable parameters (Table 2), so the overhead caused by copying data to and from the GPU memory may surpass GPU computation speedup. Also, analyzed RNN models use recurrent dropout, so a general GPU kernel is used instead of an optimized NVIDIA CUDA Deep Neural Network library (cuDNN) implementation [23].

#### 3.5.2. Incorrect Prediction Statistics

Out of 28,955 court rulings, 28,628 were categorized correctly by all three models (98.87%). Only 68 rulings were miscategorized by all three models (0.23%). Department labels for the latter are presented in Figure 10.

#### 3.5.3. Incorrect Prediction Analysis

An analysis of wrongly classified rulings reveals that most of them were not apparent—for example, some cases labeled as Civil concerned ill treatment in prison. Models often classify such cases as Labor, probably because many Labor cases concern ill treatment in the workplace. The most problematic classification is between Civil and Economic rulings. Many cases, for example, breach of contract, are labeled as Civil or Economic, depending on the status of the parties (human or company/corporation). Factors that hinder such recognition include anonymization of the parties and the frequent use of vocabulary, such as *plaintiff* (in Polish *powód*) or *defendant* (in Polish *pozwany*).

## 4. Discussion

Even simple neural networks can achieve high accuracy in court ruling classification. In the beginning of our work, we also tested other machine learning methods: Gradient Boosting, Random Forest, and SVM, but we obtained significantly worse results and did not study these models further.

General language embeddings seem to be feasible in this task. However, the proper selection of such embedding matrices has a significant impact on the performance of the model. The results are consistent with the findings by Mykowiecka et al. [3], as embeddings trained using Skip-Gram and Negative Sampling yield the best results.

Additional input in the form of the number of references to various legal codes neither improves nor worsens the model accuracy. However, it accelerates the training process. Such additional features might be useful when the number of training samples is small or when the task is more complex.

We categorized rulings into four categories based on the department, as it was present in all records and was relatively easy to reduce into several categories. As further work, one could use the topic metric to generate more labels. However, it is present in 75% of records and has over 7000 unique values (Table 1), so the generated dataset would be smaller and much more difficult to prepare.

Table 14 presents the results of legal document or sentence classification in related papers. It is difficult to compare the final results, as there are great differences in input document type, language, classification type, training set size, and the number of classes. On the other hand, the typically high $F_{1}$-scores achieved by such varied cases shows that machine learning can be successfully applied in legal document classification.

The output of our method (classification results) is available in the metadata in each record in the analyzed dataset. We do not use this data, so our algorithm can check if the proper court department proceeded with the case. Moreover, our algorithm can be used to automate choosing the department when a case is reviewed, for example, by a higher court. If we had used the department name from the metadata, the problem presented in the paper could be solved by a set of rules and would not need machine learning models.

Undoubtedly, the automatic classification of documents is useful in the legal domain. The volume of such documents is already high and is constantly growing. Machine-learning-based information systems could assist law practitioners in extracting specific information from documents and thus greatly facilitate their daily work.

## Figures and Tables

**Figure 1 sensors-22-02137-f001:**
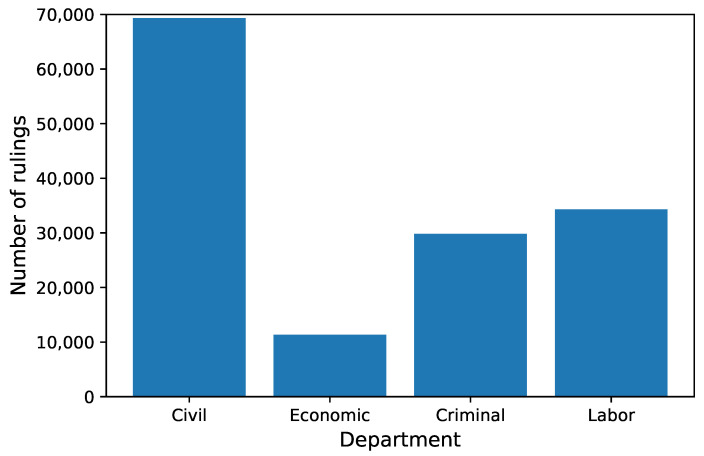
Number of rulings in each department.

**Figure 2 sensors-22-02137-f002:**
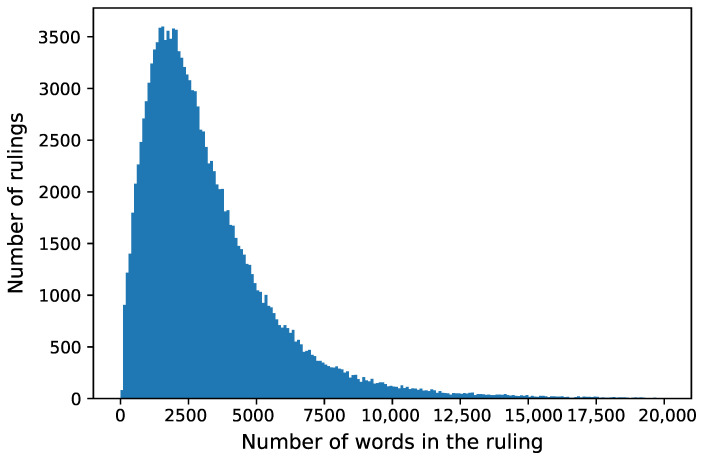
Length of rulings.

**Figure 3 sensors-22-02137-f003:**
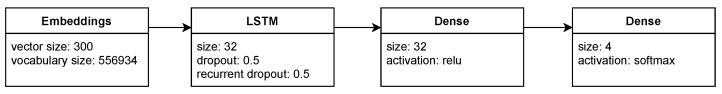
Baseline LSTM model.

**Figure 4 sensors-22-02137-f004:**
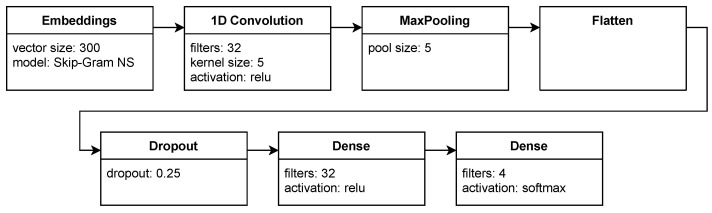
Model with a Convolutional layer.

**Figure 5 sensors-22-02137-f005:**
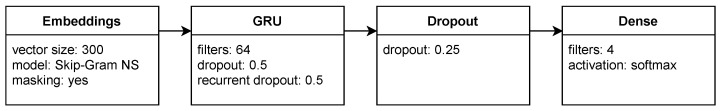
Model with a GRU layer.

**Figure 6 sensors-22-02137-f006:**
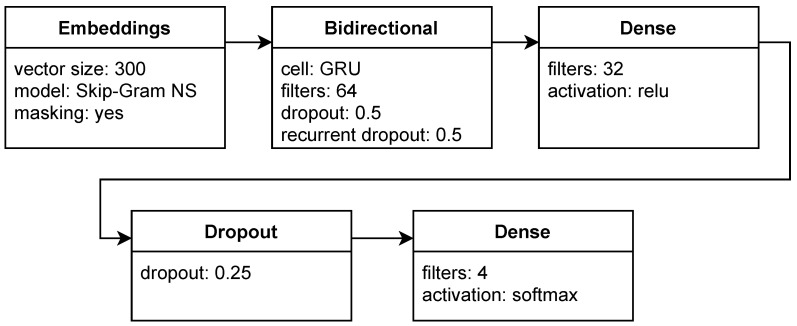
Model with a Bidirectional GRU and dense layers.

**Figure 7 sensors-22-02137-f007:**
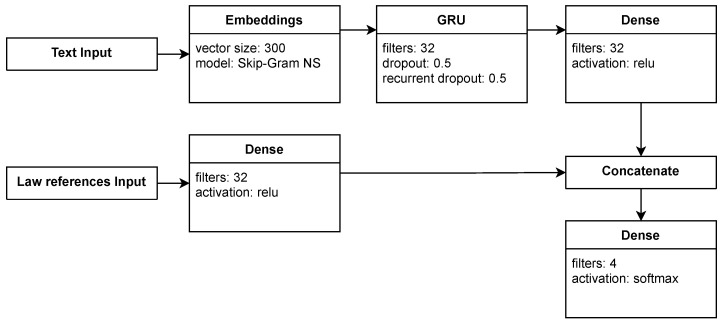
Model *gru-dense-law* (depicted in Table 2) with two inputs: text and legal codes references.

**Figure 8 sensors-22-02137-f008:**
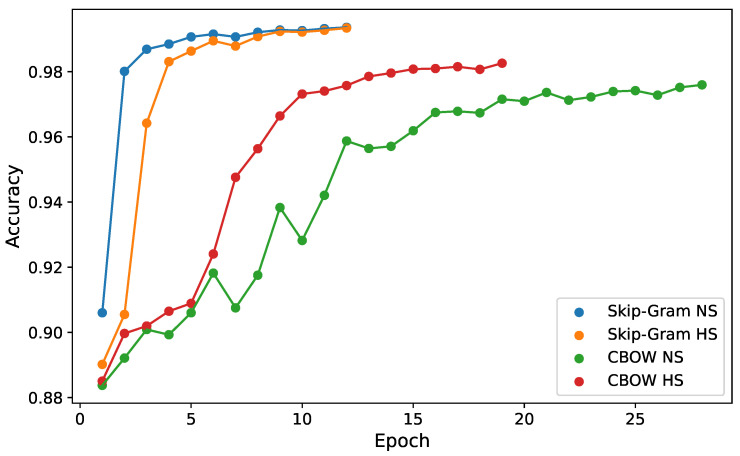
Training history accuracy for embedding matrix selection.

**Figure 9 sensors-22-02137-f009:**
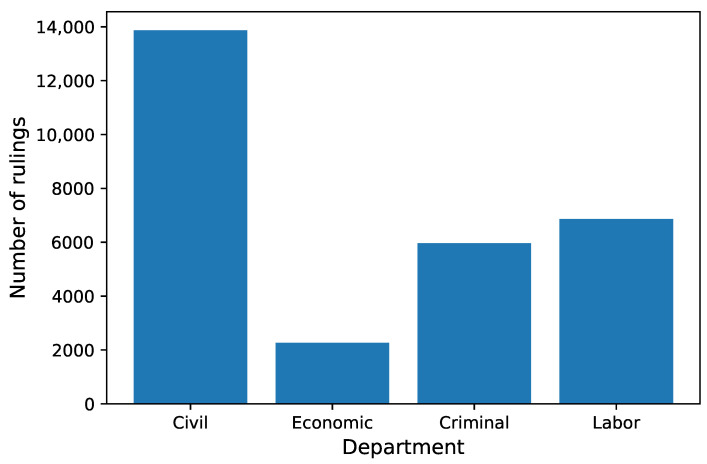
Number of rulings in each department in the test set.

**Figure 10 sensors-22-02137-f010:**
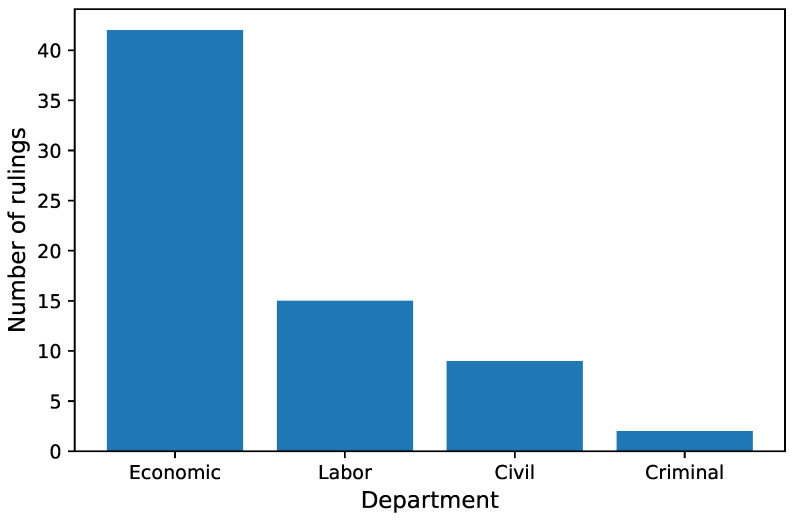
Number of rulings that were incorrectly predicted by all three models.

**Table 1 sensors-22-02137-t001:** Court ruling metrics statistics.

Metric	Coverage	Unique Values
Case ID	100%	144,784
Court ID	100%	253
Title	100%	61,321
Signature	100%	118,619
Court	100%	253
Department	100%	136
Date	100%	2827
Judge	88%	7274
Topic	74%	7037
Legal basis	75%	57,241
Thesis	9%	10,388

**Table 2 sensors-22-02137-t002:** Brief description of analyzed models.

Model	Variant	Description	Trainable Parameters
RNN + dense	*lstm-dense-skipg-ns*	The baseline model with embeddings trained on Skip-Gram and Negative Sampling (Figure 3)	43,812
	*lstm-dense-cbow-hs*	The baseline model with embeddings trained on CBOW and Hierarchical Softmax	43,812
	*lstm-dense-cbow-ns*	The baseline model with embeddings trained on CBOW and Negative Sampling	43,812
	*lstm-dense-skipg-hs*	The baseline model with embeddings trained on Skip-Gram and Hierarchical Softmax	43,812
	*lstm-dense-100*	The baseline models with smaller embeddings (100 dimensional vector instead of the default 300 dimensional one)—Skip-Gram and NS	43,812
	*gru-dense*	Similar to the baseline model but with GRU cell	33,252
	*bd-gru-dense*	Similar to the baseline model but with Bidirectional recurrent layer	66,340
Conv + dense	*conv-max-dense*	Convolutional model with max pooling and hidden dense layer (Figure 4)	181,572
	*conv-avg-dense*	Convolutional model with average pooling and hidden dense layer (Figure 4)	181,572
RNN	*lstm*	Simple model with recurrent layer with LSTM cells and no dense hidden layer	42,756
	*gru*	Simple model with recurrent layer with GRU cells and no dense hidden layer	32,196
Conv	*conv*	Simple convolutional model with average pooling and no hidden dense layer	47,908
Dense	*dense*	Model with average pooling and dense layers.	9764
Legal references	*gru-dense-law*	Similar to *gru-dense* model but has additional input with legal codes that are referenced in the rulings (Figure 7)	33,892

**Table 3 sensors-22-02137-t003:** Embedding matrix comparison.

Embedding Matrix	Accuracy	$F_{1}$-Score	Epochs
CBOW NS	0.9582	0.9319	21
CBOW HS	0.9787	0.9652	18.4
Skip-Gram NS	0.9937	0.9902	13.8
SKip-Gram HS	0.9934	0.9897	14.4

**Table 4 sensors-22-02137-t004:** Results on the validation subset for various models. The best results are presented in bold.

Model	Variant	Accuracy	$F_{1}$-Score	Epochs	Training Time
RNN + dense	*lstm-dense-skipg-ns*	0.9937	0.9902	13.8	1
	*lstm-dense-cbow-hs*	0.9787	0.9652	18.4	1.34
	*lstm-dense-cbow-ns*	0.9582	0.9319	21	1.62
	*lstm-dense-skipg-hs*	0.9934	0.9897	14.4	1.06
	*lstm-dense-100*	0.9868	0.9778	14	0.34
	*gru-dense*	0.9939	0.9903	11.8	0.79
	*bd-gru-dense*	**0.9946**	**0.9914**	9.6	0.69
Conv + dense	*conv-max-dense*	0.9905	0.9851	3.8	0.06
	*conv-avg-dense*	0.9891	0.9832	3.4	0.05
RNN	*lstm*	0.9895	0.9836	7	0.47
	*gru*	0.9939	0.9902	11.4	0.75
Conv	*conv*	0.9900	0.9843	4.2	0.06
Dense	*dense*	0.9428	0.8989	14	0.06
Legal references	*gru-dense-law*	0.9940	0.9904	9.2	0.65

**Table 5 sensors-22-02137-t005:** Metrics on validation data after hyper-parameter optimization. The best results are presented in bold.

Model	Accuracy	$F_{1}$-Score	Precision	Recall
*conv-max-dense*	0.9927	0.9886	0.9904	0.9869
*gru*	**0.9950**	**0.9921**	**0.9932**	**0.9911**
*bd-gru-dense*	0.9949	0.9917	0.9927	0.9907

**Table 6 sensors-22-02137-t006:** Metrics on the test data after hyper-parameter optimization. The best results are presented in bold.

Model	Accuracy	$F_{1}$-Score	Wrong Predictions	Correct Predictions
*conv-max-dense*	0.9924	0.9878	221	28,734
*gru*	**0.9947**	**0.9915**	**153**	**28,802**
*bd-gru-dense*	0.9942	0.9906	169	28,786

**Table 7 sensors-22-02137-t007:** Confusion matrix for the *conv-max-dense* model (defined in Table 2).

	Predicted				
		Civil	Economic	Criminal	Labor
**Actual**	Civil	13,835	27	1	4
	Economic	125	2138	0	2
	Criminal	3	1	5956	1
	Labor	48	4	5	6805

**Table 8 sensors-22-02137-t008:** Confusion matrix for the *gru* model (defined in Table 2).

	Predicted				
		Civil	Economic	Criminal	Labor
**Actual**	Civil	13,817	38	0	12
	Economic	70	2193	0	2
	Criminal	3	2	5955	1
	Labor	22	2	1	6837

**Table 9 sensors-22-02137-t009:** Confusion matrix for the *bd-gru-dense* model (defined in Table 2).

	Predicted				
		Civil	Economic	Criminal	Labor
**Actual**	Civil	13,835	26	1	5
	Economic	91	2170	0	4
	Criminal	5	0	5956	0
	Labor	31	3	3	6825

**Table 10 sensors-22-02137-t010:** Metrics on resource usage during training on the CPU.

Model	Max Memory [GB]	Full Pipeline Time [s]	Training Time [s]	Training Time per Epoch [s]
*conv-max-dense*	2.55 ± 0.03	882 ± 11	641 ± 7	160 ± 2
*gru*	2.74 ± 0.02	10,927 ± 79	10,670 ± 92	889 ± 8
*bd-gru-dense*	3.29 ± 0.04	10,031 ± 79	9771 ± 79	1086 ± 9

**Table 11 sensors-22-02137-t011:** Metrics on resource usage during training on the GPU.

Model	Max Memory [GB]	Full Pipeline Time [s]	Training Time [s]	Training Time per Epoch [s]
*conv-max-dense*	4.30 ± 0.01	216 ± 2	142 ± 1	35 ± 0
*gru*	3.52 ± 0.01	18,127 ± 98	18,005 ± 100	1500 ± 8
*bd-gru-dense*	3.72 ± 0.01	27,668 ± 110	27,538 ± 107	951 ± 4

**Table 12 sensors-22-02137-t012:** Metrics on resource usage during testing on the CPU.

Model	Model Memory [MB]	Full Pipeline Time [s]	Prediction Time [s]	Prediction Time per Sample [ms]
*conv-max-dense*	643 ± 0	286 ± 6	54 ± 0	2 ± 0
*gru*	703 ± 3	339 ± 5	95 ± 1	3 ± 0
*bd-gru-dense*	841 ± 2	389 ± 12	146 ± 1	5 ± 0

**Table 13 sensors-22-02137-t013:** Metrics on resource usage during testing on the GPU.

Model	Model Memory [MB]	Full Pipeline Time [s]	Prediction Time [s]	Prediction Time per Sample [ms]
*conv-max-dense*	965 ± 0	166 ± 3	51 ± 0	2 ± 0
*gru*	1008 ± 0	166 ± 3	51 ± 0	2 ± 0
*bd-gru-dense*	1148 ± 0	194 ± 2	80 ± 1	3 ± 0

**Table 14 sensors-22-02137-t014:** Related work comparison.

Paper	Task	Algorithm	$F_{1}$-Score
Sulea et al. (2017)	French Supreme Court rulings classification into eight categories. Trained on over 120,000 rulings.	SVM Ensemble	0.965
Noguti et al. (2020)	Petitions to Public Prosecution Service in Portuguese classification into eighteen law areas. Trained on about 16,000 documents.	SVM LR CNN LSTM GRU	0.83 0.83 0.82 0.85 0.84
Waltl et al. (2019)	German Civil Code sentences classification into nine categories. Trained on 601 sentences.	SVM LR	0.83 0.81
Undavia et al. (2018)	Classification of US Supreme Court opinions into fifteen categories. Trained on over 7500 documents.	CNN	0.72
Fernandes et al. (2020)	Classification of Brazilian Appellate Court modifications of lower court decisions proposed by the upper court. 3022 documents divided into six categories.	BI LSTM + CRF BI GRU + CRF BI LSTM BI GRU CRF	0.948 0.917 0.890 0.878 0.860
Lulu et al. (2019)	U.S Securities and Exchange Commission dataset EDGAR with 5 classes and 28,445 documents.	split into chunks + SVM	0.981
this paper	Classification of Polish court rulings into four categories. Dataset consisted of 144,784 records.	*conv-dense* *gru* *bd-gru-dense*	0.988 0.992 0.991

## Data Availability

Data used in this paper were downloaded from https://orzeczenia.ms.gov.pl/ accessed on 15 January 2022.

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
