# Peer review of "Polish Court Ruling Classification Using Deep Neural Networks"

_sensors, 2022, doi:10.3390/s22062137_

Round 1

Reviewer 1 Report

This paper presents an empirical evaluation of different word embedding methods and deep neural network architectures for the classification of Polish court rulings.

The experimental setting appears correct (in particular, the subdivision of the collected data into training and testing sets, and the use of a cross-validation procedure for model evaluation and selection on training data).

I have the following main remarks that could be addressed by a minor revision, provided that remark 1 is convincingly addressed:

1) The classification task consists in predicting the kind of a given ruling, as one of the four mutually exclusive classes: Economic, Labor, Civil and Criminal. As far as I understood, the class of a ruling is strongly related to the department of the corresponding court, and the name of the department is always available for each ruling (see table 1); moreover, according to section 2.1, the true class has been determined in this work by applying a simple set of rules to the department name (basically, they consist in merging related departments). Therefore, it is not clear to me why a relatively complex machine learning approach should be useful to this specific classification task, instead of, e.g., a much simpler rule-based approach that only considers the department name (even manually defined rules, such as the ones summarized in section 2.1).

Similarly, in section 2.1 it is stated that most rulings include an introduction which contains the name of the court department, which "could greatly facilitate classification." However, for this precise reason the introduction has been removed from the text used as the classifier input, which looks very counterintuitive. It is stated that "A similar approach was applied by Sulea O. et al. [13];" however ref. 13 dealt with a different task (automatic translation of non-legal documents), and I did not find any similar operation therein.
On the other hand, additional features related to the legal codes of the legal references in the ruling text have been used, and these codes appear strongly related to the classes of interest (e.g., Civil, Criminal and Labor Codes, as stated at the end of section 2.1).
Therefore, it seems that, on the one hand, useful information for class prediction, directly available in most rulings (their introduction), has been disregarded; on the other hand, indirect information (extracted by legal references) has been added.

The issues above make the usefulness of a machine learning approach even more questionable, at least for the specific classification task considered in this work.

2) Some details on the classification task, the data set, and the experimental setting in sections 2 and 3 are reported somewhat confusedly, and should be presented in a more systematic and orderly way:

- The number and names of the considered classes are not explicitly given in section 2.1. They can only be guessed from figure 1, and from the rest of the paper.

- Details on the embedding methods and on the neural network architectures (including their parameters) are spread across the text in section 2.2, figures 3-7 and table 2; for the reader it is difficult to follow them and clearly understand all the architectures used, also taking into account that some of them are variations of others. Perhaps there are also some typos in the mentioned figures: the number of "filters" instead of the "size" (number of units) is reported for some dense layers.

- The subdivision of the collected data set into a training and testing set is not explicitly mentioned, and can only be guessed by the reader from section 2.4 (which is however devoted to the training process) and section 3.5 (where the testing set is mentioned for the first time). I suggest reporting this detail in section 2.1, including the sizes of the training and testing sets. The way in which this subdivision has been made should also be described, e.g., was it a random subdivision or a chronological one? was it stratified (i.e., by keeping identical class proportions in training and testing data)?

- Precision, recall and their combined F1-score have been used for performance evaluation (besides other metrics). However, these metrics are defined for two-class problems, whereas the considered one is a four class problem: how have these metrics been computed?

- In section 3.5.1 it is stated that the reported performance analysis has been carried out on three different runs: how were these runs performed? For instance, have different training/testing partitions been used?

3) Additional experimental comparisons against "traditional" machine learning techniques for text categorization should be considered mandatory; in particular, I would recommend using linear SVMs and TF-IDF bag-of-word features.
Moreover, whereas the analysis of incorrect predictions in sections 3.5.3. and 3.5.3 is very interesting, for completeness I would suggest adding the confusion matrix for the three considered models.

4) For several bibliographic references the arXiv version is reported, whereas a peer-reviewed version is available (often as open access, see, e.g., the DBLP data base), and should be reported instead: refs. 6-8, 12, 13, 18, 22.

Reviewer 2 Report

The paper states that crawling and collection of court decisions was performed, a clear statement on personal data handling is required (even if it merely states that downloaded decisions had already been anonymized).

A description of the particularities of the Polish language that necessitate handling, and a concrete mapping to techniques and approaches used in the submitted work is required. 

An accurate and comprehensive description of the input used (structure and content of the decisions) should be given in the paper. 

Classification is performed at a relatively coarse-grained level using only four categories; it would be highly desirable to make a more fine-grained classification, e.g. "family cases", "fraud cases" etc.; a more fine-grained classification would provide higher additional value.

The authors state that introductions were deleted in order to make the classification task harder. However, use cases of need for classification of legal documents that are originally lacking the introductory part are not given; listing some use cases that correspond to this task would reinforce the practical value of the presented work.

The authors state that some particular embeddings were used, the properties of these embeddings should be described.

The authors state that the Adam optimization algorithm was used, however no rationale for this selection was presented (including other candidates and their properties). This also holds for the choice of the loss function.

The word "Cywilny"  is not understandable for non-Polish speakers, its meaning should be given.

Author Response

Plase see the attachment.

Round 2

Reviewer 1 Report

My remarks have been satisfactorily addressed, although for completeness also experimental results attained using alternative text categorization methods (now just mentioned in the discussion of section 4) could have been reported.